# Acute Neurochemical, Psychophysiological, and Cognitive Responses to Small-Sided Games vs. Running-Based HIIT in Young, Male Soccer Players

**DOI:** 10.3390/healthcare13141738

**Published:** 2025-07-18

**Authors:** Yakup Zühtü Birinci, Serkan Pancar, Yusuf Soylu, Hüseyin Topçu, Aygül Koçyiğit, Emre Sarandöl, Hasan Şimşek, Şenay Şahin

**Affiliations:** 1Faculty of Sports Sciences, Bursa Uludağ University, Bursa 16059, Türkiye; huseyintopcu@uludag.edu.tr (H.T.); sksahin@uludag.edu.tr (Ş.Ş.); 2Faculty of Sports Sciences, Aksaray University, Aksaray 68100, Türkiye; serkanpancar@aksaray.edu.tr; 3Faculty of Sports Sciences, Tokat Gaziosmanpasa University, Tokat 60250, Türkiye; yusuf.soylu@gop.edu.tr; 4Faculty of Medicine, Bursa Uludağ University, Bursa 16059, Türkiye; aygulkocyigit@uludag.edu.tr (A.K.); sarandol@uludag.edu.tr (E.S.); 5Faculty of Medicine, Aksaray University, Aksaray 68100, Türkiye; hasansimsek@aksaray.edu.tr

**Keywords:** brain-derived neurotrophic factor, cognition, football, game-based training

## Abstract

Background: This study aimed to compare the immediate effects of small-sided games (SSGs) and running-based high-intensity interval training (HIITrb) on serum brain-derived neurotrophic factor (BDNF) levels, cognitive performance, and enjoyment in young, male soccer players. Methods: Twenty-four soccer players [age: 19.2 ± 0.8 years] completed one session each of four-a-side SSG or HIITrb in a randomized, counterbalanced, and crossover design, with a one-week washout period. Blood samples and Trail Making Tests (TMTs) A and B were measured before and after exercise. Heart rate (HR) was monitored throughout the games, and ratings of perceived exertion (RPE) and enjoyment were collected at the end of the measurements. Results: The results show no significant effects of time (*p* > 0.775), group (*p* > 0.276) or time × group interaction (*p* > 0.199) on BDNF levels. For TMT-A, the time effect (*p* = 0.866) and group effect (*p* = 0.057) were not significant; however, the time × group interaction was significant (*p* < 0.019), indicating a superior performance in the SSG compared to HIITrb. In the TMT-B, significant effects were observed for both time (*p* < 0.001) and group (*p* < 0.001), while the time × group interaction effect was not statistically significant (*p* > 0.061). Furthermore, enjoyment levels did not differ significantly between conditions (*p* = 0.976). Conclusions: These findings suggest that four-a-side SSG may enhance processing speed compared to HIITrb without changes in serum BDNF levels. Coaches may consider using 4v4 SSG formats in early training sessions or warm-ups to stimulate processing speed and mental readiness in young soccer players.

## 1. Introduction

Soccer is a physically demanding, high-intensity team sport characterized by an intermittent activity pattern, involving repeated sprints, jumps, and directional changes, interspersed with active and passive recovery [1]. It also imposes substantial demands on players’ perceptual and cognitive skills, requiring rapid, context-dependent decisions, such as dribbling, passing, and shooting, under time pressure [2,3]. As modern games have evolved into faster and more complex sports, coaches should adopt integrated training strategies to prepare players comprehensively.

High-intensity interval training (HIIT), alternating short bouts of intense exercise exceeding 90% peak heart rate (HR) with low-intensity recovery, has been widely incorporated into soccer training programs to enhance performance [4]. A meta-analysis by Kunz et al. [5] highlighted the effectiveness of different HIIT modalities, including running-based high-intensity interval training (HIITrb) and small-sided games (SSGs), a soccer-specific form of HIIT, in improving youth players’ performance. However, HIITrb alone may inadequately replicate the cognitive demands of match play, limiting essential factors, such as movement variability, action unpredictability, and perceptual-motor adaptability necessary for effective performer–environment interactions in team sports [6].

In contrast, SSGs integrate physical, technical, and tactical elements more holistically, aligning closely with the multifaceted demands of match play, and have consequently become widely preferred by coaches as effective training tools [7]. The evidence suggests that SSGs can elicit comparable improvements in physiological and physical performance outcomes, while offering additional benefits in technical–tactical development and psychophysiological responses [8]. Furthermore, SSGs, by engaging players with the ball, have been shown to enhance cognitive functions, such as decision-making, attention, and spatial awareness [9]. However, despite the well-documented and extensive body of knowledge, no study has yet explored the neurobiological mechanisms underlying the additional effects of SSGs, leaving a critical gap in understanding their comprehensive benefits.

Brain-derived neurotrophic factor (BDNF), released into the bloodstream during physical activity, is a key mediator of exercise-induced cognitive benefits and neuroprotective effects, including the inhibition of neurodegeneration and enhancement of hippocampal plasticity through neurogenesis, axonal growth, and synaptogenesis [10,11]. Exercise-induced structural adaptations mediated by the BDNF have also led to its recognition as a reliable marker of brain function during the lifespan of an individual [12,13]. Building on these general findings, recent studies have also proposed the BDNF as a promising biomarker for monitoring training load [14] and enhancing cognitive performance, specifically in athletes [15]. Furthermore, the BDNF has been linked to psychological improvements in athletic populations, including anger regulation, mental fatigue resistance, stress coping, competitive anxiety reduction, and increased mental toughness [16,17].

Although progress has been made in identifying the structural brain changes induced by exercise, the specific exercise modalities that trigger the underlying neurobiological mechanisms related to cognitive enhancement remain unclear. The evidence indicates that exercise stimulates blood-borne BDNF release, with aerobic [18], resistance [19], concurrent [20], HIITrb [21], and soccer-specific [14] training eliciting various responses. Additionally, exercises that impose high neuromuscular and cognitive demands have been shown to result in higher serum BDNF levels [22,23,24], although our previous findings did not support this trend [25]. Some studies also suggested that exercise modalities requiring a higher cognitive load and motor coordination may lead to an enhanced cognitive performance [26,27]. Nonetheless, the existing research is yet to determine the optimal exercise modality for enhancing cognitive performance via BDNF stimulation.

To the best of our knowledge, this study is the first to comprehensively investigate the acute effects of SSGs and HIITrb on neurochemical, cognitive, and psychophysiological responses in soccer players. By addressing this gap in the literature, this study offers empirical evidence on how different HIIT modalities influence neurocognitive and psychophysiological adaptations in soccer through BDNF-mediated mechanisms. Therefore, this study aimed to investigate and compare the immediate effects of SSGs and HIITrb drills on serum BDNF levels, cognitive performance, and enjoyment in young, male soccer players. We hypothesized that SSGs would elicit greater acute increases in serum BDNF levels, higher enjoyment, and enhanced cognitive performance than HIITrb.

## 2. Materials and Methods

### 2.1. Subjects

G*Power 3.1 software (version 3.1.9.7, University of Düsseldorf, Düsseldorf, Germany) was used to determine the sample size using repeated-measures ANOVA (within-subjects). Assuming a moderate effect size (f = 0.25), a significance level of 5% (α = 0.05), a statistical power of 80% (1 − β = 0.80), a correlation of 0.6 between repeated measures, and sphericity (ε = 1), the minimum sample size required was 23 participants. Twenty-four semi-professional, male, youth soccer players [age: 19.2 ± 0.8 years; height: 181.5 ± 5.8 cm; body mass: 78.4 ± 10.3 kg; body fat percentage: 11.4 ± 6.6%; maximal oxygen consumption (VO2max): 51.5 ± 3.2 mL·kg^−1^·min^−1^] voluntarily participated in this study during the 2024–2025 competitive season. All participants (sports background of at least 5 years and a training workload of at least 4 d/w) were chosen from the same team competing in the Regional Amateur League, which represents a competitive semi-professional level in the national football system. Participants were recruited after meeting the following inclusion criteria: official medical clearance, no recent severe lower-extremity injury (>12 months), and no recent use of medications or supplements that could enhance anaerobic or aerobic performance. The exclusion criteria were smoking, drug and alcohol use, taking psychoactive drugs, a history of depression or neurological illness or injury, and neuromotor or musculoskeletal disorders. Before signing the informed consent form, the participants were informed of the research procedures, benefits, and risks.

### 2.2. Study Design

A randomized crossover design (two experimental conditions) was used to investigate the acute effects of SSGs and HIITrb training drills on serum BDNF levels and cognitive performance in male soccer players. Teams were formed based on the coach’s assessment of the players’ technical, tactical, and physical abilities [28]. To minimize differences in fitness levels, player rankings from the Yo-Yo Intermittent Recovery Test Level 1 (YYIRT-1) were also taken into account. The SSGs and HIITrb sessions were conducted in the morning (10.00–12.00) to have similar chronobiological characteristics on an artificial-grass pitch during the competitive season training period. All players continued their routine training program, which involved five training sessions and one match per week. Players were advised to avoid strenuous exercise within 48 h, be prohibited from drinking alcohol or coffee within 24 h, and have at least 7–8 h of sleep before the intervention. During this study, the players were instructed to maintain their normal and habitual lifestyle habits. The temperature (14–15 °C) and humidity (30–35%) levels were similar during the interventions.

### 2.3. Procedures

This study analyzed the physiological, psychophysiological, and neurochemical responses and cognitive function performance of male soccer players during both HIITrb and SSGs. The intervention consisted of three visits on separate days, with a 1-week interval between each visit. The intervention procedure is summarized in Figure 1.

During the first visit, body weight and body fat percentage were measured using bioelectrical impedance measurements (BC-418; Tanita, Tokyo, Japan). After anthropometric measurements, resting HR was recorded for each player. To mitigate potential learning effects, all participants underwent a familiarization session with Trail Making Tests (TMTs) A and B. Finally, the YYIRT-1 test was conducted to determine the aerobic capacity, maximal aerobic speed (MAS), and maximal heart rate (HRmax).

During the two subsequent visits, each participant completed both HIITrb and SSG training sessions, with one session (HIITrb or SSG) completed during visit 2 and the other during visit 3. The order of sessions was randomized and counterbalanced, such that half of the participants completed HIITrb first and the other half completed the SSG first.

Blood samples were collected and cognitive performance tests were conducted before and after each training session. Before each intervention session, participants completed a brief practice trial of both TMT-A and TMT-B to reinforce task familiarity and reduce practice-related performance gains unrelated to the intervention. The HR was monitored throughout each session using a Polar V800 device (Polar OY; Kempele, Finland). Each exercise session began with a 20 min warm-up that included jogging, dynamic stretching, and football-specific movements. This was followed by four sets of 4 min exercise bouts (either a 4-a-side SSG or HIITrb), each separated by a 3 min passive rest interval. All exercise sessions lasted ~45 min. All sessions were conducted under the supervision of an expert strength and conditioning coach, to ensure proper execution and adherence to the protocol. The details of all exercise sessions are provided below.

Additionally, following each exercise session, ratings of perceived exertion (RPE) and Exercise Enjoyment Scale (EES) scores were used to assess the players’ psychophysiological responses. Water intake was permitted during all the recovery periods throughout the training sessions.

#### 2.3.1. Small-Sided Game

In this study, we conducted 4-a-side SSG to foster environments recommended by a recent review [29], suggesting that game formats with fewer players may offer a more cognitively demanding structure, including creativity, exploratory behaviors, and decision-making challenges for players. The SSG was performed on an outdoor artificial-grass field (25 m × 35 m) using small goals without goalkeepers. The duration of the SSG was carefully standardized, consisting of four 4 min bouts with 3 min passive recovery intervals between bouts, as outlined in previous studies [30]. Players were instructed to exert maximal effort during gameplay and received verbal encouragement, which excluded feedback on their technical and tactical performance [31]. To ensure uninterrupted play and maximize effective playing time, multiple balls were placed along the sidelines, and two coaches stationed around the field promptly provided replacement balls when required [32]. The SSG was free-flowing without imposed constraints, such as touch limitations, mandatory passing, or predefined scoring conditions (Table 1).

#### 2.3.2. Running-Based High-Intensity Interval Training

HIITrb was conducted in an outdoor artificial-grass field. Players covered a predetermined distance during 15 s intervals, followed by 15 s of passive rest before starting the next interval, running in the opposite direction. The distance was individualized for each player based on their MAS derived from the YYIRT-1 test and was set at 110% of their MAS [33]. This protocol consisted of four 4 min bouts, each separated by 3 min of passive rest (Table 2).

### 2.4. Measurements

#### 2.4.1. Anthropometric Measurements

Before breakfast, the players’ weight and height were measured using a body composition analyzer (BC-418MA, Tanita Corp., Tokyo, Japan). Using the bioelectrical impedance method, this analyzer uses multiple frequencies (ranging from 1 to 50 kHz) to perform detailed body composition measurements.

#### 2.4.2. Yo-Yo Intermittent Recovery Level 1 Test

The participants performed YYIRT-1, a reliable, progressively challenging, and acoustically guided evaluation to assess aerobic capacity, following the protocol outlined by Bangsbo et al. [34]. The HR of the participants was monitored throughout the test using a Polar V800 device (Polar OY, Kempele, Finland). The peak HR observed during YYIRTL-1 was recorded as the HRmax. Upon completion of the test, the researchers estimated VO2max using a predetermined formula:VO2max = 36.4 + (0.0084 × covered distance in YYIRT-level 1).

#### 2.4.3. Measure of Exercise Intensity

To objectively assess exercise intensity, HR (recorded at 5 s intervals) was continuously recorded using heart rate monitors (Polar V-800; Polar Electro OY, Kempele, Finland) during the sessions. In addition, participants used a Polar H10 chest strap for HR measurements. The average HR values at each repetition time were evaluated during the data processing phase. HR data are presented as percentages of the maximum HR (%HRmax) and HR reserve (%HRreserve). The mean HR (HRmean) was calculated for each training session (HIITrb and SSG). The %HRmax for each training type was determined using the following formula:%HRmax = (HRmean/HRmax) × 100

The %HRreserve was calculated using the following formula:%HRreserve = (HRmean − HRrest)/(HRmax − HRrest) × 100

#### 2.4.4. Assessment of Cognitive Function Performance

Cognitive performance was evaluated using TMT-A and TMT-B and has been validated by Türkeş et al. [35] in Turkish young adults (20–49 years old). TMT evaluates attention, visual scanning, visuomotor speed, and cognitive flexibility. While TMT-A and TMT-B are correlated, TMT-B is considered more sensitive to executive functions, particularly cognitive flexibility [36]. The test was administered in a paper-and-pencil format and included two separate parts. In condition A, the participants were instructed to connect 25 encircled numbers (1 to 25) in ascending numerical order (i.e., 1–2–3…) as quickly and accurately as possible without lifting the pen from the paper. In condition B, the task required alternating numbers and letters in an ascending alphanumeric sequence (i.e., 1–A–2–B–3–C… up to 13–L). Prior to each part, the participants were provided standardized instructions and a brief practice trial to ensure task comprehension. The main outcome measure for both tests was the total time (in seconds) required to correctly complete each sequence. Any errors were immediately indicated by the examiner and corrected by the participant during the trial; the correction time was included in the total completion time.

#### 2.4.5. Psychophysiological Responses

The EES was developed to assess players’ perceived enjoyment levels during and after an activity [37]. The Turkish adaptation of the scale was developed by Soylu et al. [38]. It consists of eight items scored on a 7-point bipolar Likert-type scale. The maximum score obtained from the scale was 56 and the minimum score was 8. RPE was measured using Borg’s 6–20 scale [39].

#### 2.4.6. Biochemical Analysis

Blood samples (8 mL) were collected from the antecubital vein before and immediately after exercise while seated. After clotting at room temperature for 30 min, the samples were centrifuged at 3000× *g* for 15 min, and the serum was frozen at −80 °C for later analysis. Serum BDNF levels were measured using enzyme-linked immunosorbent assay methods with an ELISA Kit (Cloud Clone, USCNK, Wuhan, China). To measure blood lactate concentration, a Super GL2 analyzer (Müller Gerätebau GmbH, Freital, Germany) was utilized. The analysis employed an enzymatic–amperometric electrochemical technique. The lactate analyzer was calibrated before each analysis session for every participant according to the manufacturer’s recommendations. Blood samples were decoded only after completion of the analysis.

### 2.5. Statistical Analyses

Data are presented as mean ± standard deviation. IBM SPSS Statistics (version 27.0, SPSS Inc., Chicago, IL, USA) was used for all the analyses. Normality was evaluated using the Shapiro–Wilk test. Paired-sample *t*-tests were used to compare within-subject responses between the SSG and HIITrb conditions for acute psychophysiological outcomes (HRmean, RPE, and enjoyment). Two-way repeated-measures ANOVA was used to analyze BDNF, lactate, TMT-A, and TMT-B performance. The Mauchly test confirmed the assumption of sphericity, and the Greenhouse–Geisser correction was used when this assumption was violated. Bonferroni-corrected tests were used for post hoc comparisons. Partial eta squared ($\eta_{p}^{2}$) values were reported as effect size (small: 0.01 ≤ $\eta_{p}^{2}$) < 0.06; (medium: 0.06 ≤ $\eta_{p}^{2}$) < 0.14; (large: $\eta_{p}^{2}$) ≥ 0.14) [40]. Statistical significance was set at *p* < 0.05. All results are reported as the mean ± standard deviation and 95% confidence intervals. All graphs were generated using estimated marginal means, and error bars in the figures represent 95% confidence intervals, unless otherwise specified. Visualization was performed using JAMOVI (Version 2.3) and JASP (Version 0.17.1).

## 3. Results

In the following section, statistical findings for both the SSG and HIITrb protocols are presented (Table 3 and Table 4).

In Figure 2, the BDNF results show that the time effect is not statistically significant [F(1, 46) = 0.083, *p* > 0.775, $\eta_{p}^{2}$ = 0.002]. The group effect was also not significant [F(1, 46) = 1.21, *p* > 0.276, $\eta_{p}^{2}$ = 0.026]. Time × group interaction was not significant [F(1, 46) = 1.73, *p* > 0.199, $\eta_{p}^{2}$ = 0.036].

The lactate results demonstrate that the time effect is statistically significant [F(1, 46) = 1286.86, *p* < 0.001, $\eta_{p}^{2}$ = 0.965]. The group effect was not significant [F(1, 46) = 2.98, *p* < 0.091, $\eta_{p}^{2}$ = 0.061]. Time × group interaction was also not significant [F(1, 46) = 0.084, *p* < 0.773, $\eta_{p}^{2}$ = 0.002].

The TMT-A results reveal that there is no significant main effect of time [F(1, 46) = 0.0286, *p* = 0.866, $\eta_{p}^{2}$ = 0.001]. The main effect of group was not significant [F(1, 46) = 3.80, *p* = 0.057, $\eta_{p}^{2}$ = 0.076]. However, the time × group interaction was significant [F(1, 46) = 11.73, *p* = 0.019, $\eta_{p}^{2}$ = 0.203], indicating that changes in performance over time differed between groups. Post hoc comparisons show that the SSG group showed an improvement in performance from pre-test to post-test (decrease from 12.8 s to 11.4 s), whereas the HIITrb group showed a small decrease in performance (increase from 13.2 s to 14.5 s).

A two-way repeated measures ANOVA revealed a significant main effect of time on TMT-B performance [(F(1, 46) = 24.30, *p* < 0.00, $\eta_{p}^{2}$ = 0.346], indicating improved scores from the pre- to post-test across both groups. There is also a significant main effect of group [F(1, 46) = 18.38, *p* < 0.001, $\eta_{p}^{2}$ = 0.286]. The time × group interaction was not significant [F(1, 46) = 3.69, *p* > 0.061, $\eta_{p}^{2}$ = 0.074].

In Figure 3, the *t*-test results show that there is no statistically significant difference in HRmean (t = −0.087; *p* > 0.931), RPE (t = −0.882; *p* > 0.382), and enjoyment (t = 0.034; *p* > 0.973) between the SSG and HIITrb groups.

## 4. Discussion

The present study examined the immediate effects of a four-a-side SSG and HIITrb on serum BDNF levels, cognitive performance, and enjoyment in young, male soccer players during the competitive season. A key novelty of this research is the assessment of neurochemical mechanisms, particularly BDNF, which may contribute to cognitive function differences between HIITrb and SSGs. Our findings indicate that HIITrb (four 4 min bouts of intermittent running at 110% of MAS, with 15-s work intervals followed by 15 s of passive rest, and 3 min rest periods between bouts) and a SSG (4-a-side SSG: four, 4 min bouts with 3 min passive recovery intervals) sessions induced similar physiological (HRmean, blood lactate), neurochemical (serum BDNF), and psychophysiological (EES, RPE) responses. However, the SSG resulted in a greater improvement in cognitive performance (TMT-A) than HIITrb.

### 4.1. Serum BDNF Levels

Previous evidence suggests that HIIT can enhance brain function primarily by increasing neurotrophin release [12]. Supporting this, previous research, including systematic reviews and meta-analyses, have shown elevated serum BDNF levels following a single HIIT session [21,41,42,43]. This response may be driven by physiological mechanisms, such as increased hydrogen peroxide (H_2_O_2_) and tumor necrosis factor-alpha (TNF-α), which mediate oxidative stress and inflammatory signaling in the brain [44]. These molecules activate nuclear factor-κB (NF-κB), which facilitates BDNF gene expression. Additionally, HIIT may upregulate peroxisome proliferator-activated receptor gamma coactivator 1-alpha (PGC-1α) via the activation of key signaling pathways (kinases and phosphatases), which in turn promotes fibronectin type III domain-containing protein 5 (FNDC5) expression, a known upstream regulator of BDNF [45]. Lactate accumulation during HIIT has also been proposed to stimulate BDNF synthesis by enhancing N-methyl-D-aspartate (NMDA) receptor activation and intracellular Ca^2+^ signaling [46]. However, the present study did not observe an increase in serum BDNF levels following acute HIIT in soccer players, suggesting possible population-specific or contextual differences.

There are a limited number of studies that have examined the effects of HIIT on BDNF levels in athletic populations. While some studies reported increased BDNF levels following various HIIT protocols [47,48,49], a meta-analysis by García-Suárez et al. [43] suggested that HIIT is often perceived as strenuous by athletes, potentially elevating systemic cortisol levels. Supporting this, higher cortisol levels have also been reported in semi-elite soccer players during the competitive season [50]. Notably, in-season periods expose athletes to cumulative physical and cognitive stress due to intensified training regimens and frequent match play, which is typically accompanied by insufficient recovery periods. A study by Andrzejewski et al. [14] that monitored changes in biomarkers, such as BDNF, across different phases of the football season reported a significant reduction in serum BDNF levels during the competitive period in youth soccer players, even though the overall training load was lower than in the preparatory phase. In the present study, BDNF levels were assessed during the competitive period in subprofessional players. Taken together, the lack of change in BDNF levels following HIITrb may be attributed to elevated stress hormone activity, particularly cortisol, which is known to suppress BDNF synthesis. As a possible mechanism, Herhaus et al. [51] described an antagonistic interplay between cortisol, a major glucocorticoid hormone in humans, and BDNF, mediated via the hypothalamic–pituitary–adrenal (HPA) axis in response to both acute and chronic stress.

Although the research on the impact of HIITrb on BDNF is expanding, studies focused on soccer-specific exercises are still limited and have shown inconsistent results. For instance, Yang et al. [52] reported a significant acute increase in serum BDNF levels following a 90 min soccer match in adolescent males. However, the SSG format used in the present study differs substantially from full match-play in terms of physiological, tactical, and technical demands, which may influence neurochemical responses. To date, only one study has specifically investigated the effects of SSGs on BDNF expression. Consistent with our findings, Williams et al. [53] reported no change in the BDNF following a soccer-specific training session—comprising a 5 min warm-up, 25 min of skill-based drills, and 30 min of 5-a-side SSG—in adolescent players, citing potentially insufficient intensity (%HRmax: 75 ± 8 beats·min^−1^). Similarly, a systematic review by Gutierrez-Vargas et al. [15] concluded that the current evidence is insufficient to draw firm conclusions regarding the effects of soccer-related exercise on BDNF response. Given the complex, intermittent, and multifactorial nature of soccer, further research is needed to better understand its neurotrophic effects.

One of the main hypotheses of this study was that an exercise modality with higher cognitive demands, such as a SSG, would elicit a greater increase in BDNF levels compared to HIITrb of a similar intensity and volume. However, our results do not support this, as no significant difference in BDNF levels was observed between the two modalities. Open-skill exercises (OSEs), which are performed in unpredictable, externally paced environments and require continuous perceptual monitoring, rapid decision-making, and motor adaptability, may produce greater neurocognitive engagement and BDNF upregulation than closed-skill formats. In contrast, closed-skill exercises (CSEs) are characterized by consistent and repetitive motor patterns performed in predictable settings, imposing relatively low cognitive demands, such as running on a treadmill or cycling. Previous studies have shown that OSEs, such as badminton [22] and dance [23], lead to greater increases in serum BDNF levels compared to CSEs. This effect is often attributed to the complex nature of OSEs, which simultaneously engage cognitive and physical processes, thereby potentially inducing synergistic stimulation of BDNF production. However, other studies have also reported acute elevations in BDNF levels following activities such as table tennis [25], badminton [54], and fencing [55]; however, these studies did not observe significant differences when compared with CSE modalities. In our study, although the SSG involved complex perceptual and motor demands characteristic of OSEs, this did not lead to superior BDNF responses compared to HIITrb. Thus, while OSE-like SSGs hold theoretical promise, further research with refined designs and more sensitive biomarker tracking is needed to clarify their neurochemical impact. These inconsistent findings may be attributed to methodological heterogeneity. Key sources of variation include exercise variables, cognitive task types, training status of participants, and the timing of BDNF assessments.

The lack of significant differences in BDNF levels between HIITrb and SSGs may be explained by their similar exercise intensities, as indicated by comparable lactate responses. Recent discoveries indicate that lactate may play a more crucial role in BDNF release than exercise modality per se [56,57]. Another possible explanation may lie in the adaptive physiological responses of trained individuals. In active populations, circulating BDNF may be rapidly utilized for muscle repair during or after exercise rather than remaining elevated in the serum [58]. Our results show that neither the SSG nor HIITrb protocols significantly changed serum BDNF levels. This null finding occurred despite the sufficient power analysis and sample size to detect moderate effects. This aligns with the contradictory results in the literature [43,57]. Reycraft et al. [57] noted that BDNF increases after HIIT and varies with exercise intensity and recovery. The absence of a BDNF response could be due to participants’ high basal cortisol levels during the competition season or rapid neurotrophin utilization in peripheral tissues, although further studies are needed to test these hypotheses. Although direct evidence is still limited, we can speculate on the putative wound-healing function of BDNF. It is plausible that circulating BDNF was directed toward tissue recovery functions, thus explaining the absence of measurable increases post-exercise. This perspective aligns with the notion that the tendency for decreased BDNF levels following SSGs may be related to greater external load, such as acceleration, deceleration, changes in direction, and jumps, which are known to induce muscle damage.

### 4.2. Cognitive Functions Performance

In the present study, SSGs led to improved TMT-A performance, as indicated by faster completion times. These findings are consistent with the growing body of evidence in the exercise-cognition research, which suggests that even a single session of exercise can transiently enhance performance in specific tests assessing various cognitive domains. Previous studies have similarly reported that acute HIIT can improve cognitive performance in young adults [59,60].

To the best of our knowledge, few studies have explored the cognitive effects of soccer-specific exercises. Lind et al. [61] reported that 3-a-side SSG performed at high intensity (70–100% HRreserve) enhanced inhibitory control and attention-related EEG responses in children, compared to moderate intensity, though no improvement was found in declarative memory across any condition. Similarly, Alesi et al. [62] observed improvements in visual discrimination and selective attention following a six-month structured football program, which consisted of individual skill drills and 3- to 5-a-side SSGs performed twice per week for 75 min in sedentary children. Hammami et al. [63] found that both SSGs (two sets of 5v5 or 6v6 formats) and sprint exercises acutely improved attention (d2-test) in adolescents. Skala and Zemkova [64] reported that while youth players responded faster in a Go/No-Go task after a 30 min (six 4 min bouts) 4-a-side SSG, errors of commission also increased, suggesting that fatigue may impair decision-making. Given the similarity of protocols, this may explain why our study observed improvements in TMT-A (processing speed) but not in TMT-B, which more specifically reflects executive control and decision-making.

While significant improvements occurred in TMT-A scores after the SSG, TMT-B performance showed no changes. This difference may stem from the cognitive demands of the tasks. TMT-A measures processing speed and attention, which appear more responsive to aerobic and game-based exercises. TMT-B requires higher executive functioning, such as cognitive flexibility. Studies have indicated that complex cognitive tasks may require longer exercise exposure for improvement [65,66]. The lack of TMT-B changes likely reflects the limitations of acute interventions in targeting higher-order cognition.

Our findings, which demonstrate improvements in cognitive performance (TMT-A) following SSGs, may be partly attributed to the sport-specific cognitive demands inherent in soccer. The dynamic and unpredictable nature of gameplay requires players to engage in continuous perceptual monitoring, such as interpreting environmental cues, such as the positioning of teammates and opponents, alongside rapid decision-making and accurate motor execution under pressure, such as passes, dribbles, or shots. These tasks depend heavily on executive function, including selective attention, working memory, and cognitive flexibility. Importantly, TMT-A specifically captures aspects such as processing speed, visual scanning, graphomotor speed, and visuomotor processing speed, skills that are particularly relevant to football-specific cognitive performance. While TMT-B did not show significant changes and thus limits broader interpretations regarding higher-order executive functions, this distinction is important to avoid overgeneralization and accurately contextualize the cognitive benefits observed.

Given that SSGs retain these core cognitive demands within a condensed and intensive format, they may serve as a particularly effective stimulus for eliciting acute cognitive benefits beyond those observed through general physical activity alone. Although the previous research suggests that elevated lactate levels may mediate executive function improvements following HIIT [67], both exercise protocols in the present study elicited comparable lactate responses; however, only the SSG enhanced cognitive performance. This finding indicates that metabolic stress, such as lactate, alone may not account for the observed cognitive benefits. Instead, the neural mechanisms may play a more prominent role. Prior studies have shown that acute HIIT may facilitate executive function through improved neuroelectric efficiency, faster stimulus processing [68], and increased prefrontal cerebral oxygenation [69]. While our study did not observe cognitive improvements linked to neurochemical markers, such as BDNF or lactate, it is plausible that other mechanisms, such as molecular and cellular signaling (e.g., irisin, insulin-like growth factor-1 (IGF-I), vascular endothelial growth factor (VEGF), interleukin-6 (IL-6), and neuroplasticity, or socioemotional factors) contribute to the observed cognitive benefits.

### 4.3. Enjoyment Level

Another interesting finding of our study was that no significant difference was found between SSGs and HIITrb, as measured by the EES scores. Previous research frequently highlights that SSGs are more enjoyable and have positive psychological effects on players [8,30,70,71], while others report contradictory findings [72,73]. A SSG creates a training environment that mirrors real-game situations, with more frequent ball touches, decision-making opportunities, and greater overall involvement, which can enhance intrinsic motivation and enjoyment [70]. Thus, SSGs are preferable to traditional drills, such as HIITrb, which often lack the engagement and motivational factors of actual gameplay [30].

However, our findings do not support the assumption that these factors contribute to greater enjoyment in SSGs. Similarly, Hammami et al. [73] found no significant differences in EES scores between repeated sprint sessions and three-a-side SSG in an untrained youth population, with both modalities resulting in relatively low perceived enjoyment. A recent review by Clemente [72] suggested that while SSGs are generally superior to HIITrb in enhancing player enjoyment, factors such as training format, coaching behavior, and individual preferences may influence enjoyment levels. The author noted that the existing literature remains inconsistent and limited, making it difficult to draw definitive conclusions.

Although no significant differences were observed between the conditions in terms of enjoyment, the enjoyment construct remains a critical psychological factor for training adherence and long-term engagement. Prior research has emphasized that higher perceived enjoyment is positively associated with greater intrinsic motivation and continued participation in physical activity programs, particularly among young athletes [37,72]. According to the self-determination theory, enjoyment reflects intrinsic motivation and supports autonomy, which is essential for fostering sustained behavior [74]. Therefore, while enjoyment levels were statistically similar between SSGs and HIITrb in our sample, the maintenance of moderate-to-high enjoyment across both modalities may still have practical implications for adherence to training programs, especially in youth soccer development contexts.

The present study has several limitations. First, the sample consisted exclusively of well-trained, motivated, and healthy, young, male soccer players, which limits the generalizability of the findings to broader populations, including females, older individuals, and untrained participants. Second, serum BDNF concentrations were assessed at a single time point immediately following exercise. However, circulating BDNF levels are highly sensitive to the timing of post-exercise blood collection and can fluctuate within a short time window [75]. Third, cognitive performance was assessed solely using TMT-A and TMT-B. Although these tests are valid and widely used, prior research suggests that the type of cognitive task may significantly influence the effects of both acute and chronic exercises on cognitive outcomes [66]. Another limitation relates to the players’ habituation to both training types. Since participants were regularly exposed to SSGs and HIITrb during in-season training, enjoyment scores may have been affected by a lack of novelty. The literature suggests exercise enjoyment is shaped by perceived novelty of training stimulus [37,72]. More research is needed to explore novel training protocols to better assess enjoyment and motivational responses.

Future studies should incorporate a broader range of cognitive assessments and include additional neurocognitive biomarkers, such as irisin, IGF-1, IL-6, and VEGF, to better elucidate the mechanisms underlying exercise-induced cognitive benefits. More frequent and time-sensitive post-exercise blood sampling would also be beneficial for accurately characterizing the temporal profiles of neurotrophic responses. Future research should also aim to replicate and extend these findings in more diverse populations, including female athletes, to improve the generalizability of outcomes and explore potential sex-specific responses to exercise interventions. Furthermore, longitudinal studies tracking neurotrophic markers over extended training periods may offer deeper insights into the sustainability and long-term impact of the SSG and HIITrb protocols on neurobiological adaptations, cognitive function, and sport-specific performance outcomes.

## 5. Conclusions

The present study found that acute game-based HIIT (four-a-side SSG) elicited superior performance, as evidenced by shorter completion times in TMT-A, which assesses visual scanning, graphomotor speed, and visuomotor processing speed, than HIITrb in young, male soccer players. However, both exercise modalities failed to elicit significant improvements in TMT-B, which is considered a more sensitive measure of executive function. Moreover, BDNF levels were not influenced by either the SSG or HIITrb, and no differences were detected between exercise modalities. These findings do not support the assumption that the BDNF mediates the observed improvement in cognition. From a theoretical standpoint, the observed improvement in cognitive performance following SSGs may support the view that simultaneously engaging in cognitively demanding tasks during physical activity can enhance neurocognitive efficiency. Accordingly, coaches and practitioners should consider embedding game-based HIIT formats into training routines, as these formats not only mimic the physical demands of soccer, but also serve as an effective strategy to enhance soccer-specific cognitive skills, particularly processing speed and visual scanning.

## Figures and Tables

**Figure 1 healthcare-13-01738-f001:**
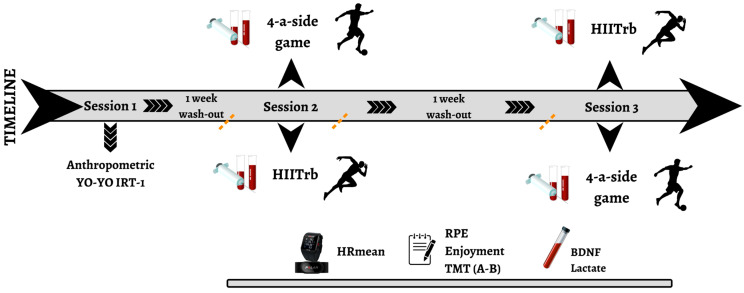
Study design. BDNF: brain-derived neurotrophic factor; HIITrb: high-intensity interval training; HRmean: mean heart rate; RPE: rating of perceived exertion; SSG: small-sided game; TMT: Trail Making Test; YYIRT-1: YO-YO Intermittent Recovery Test Level 1.

**Figure 2 healthcare-13-01738-f002:**
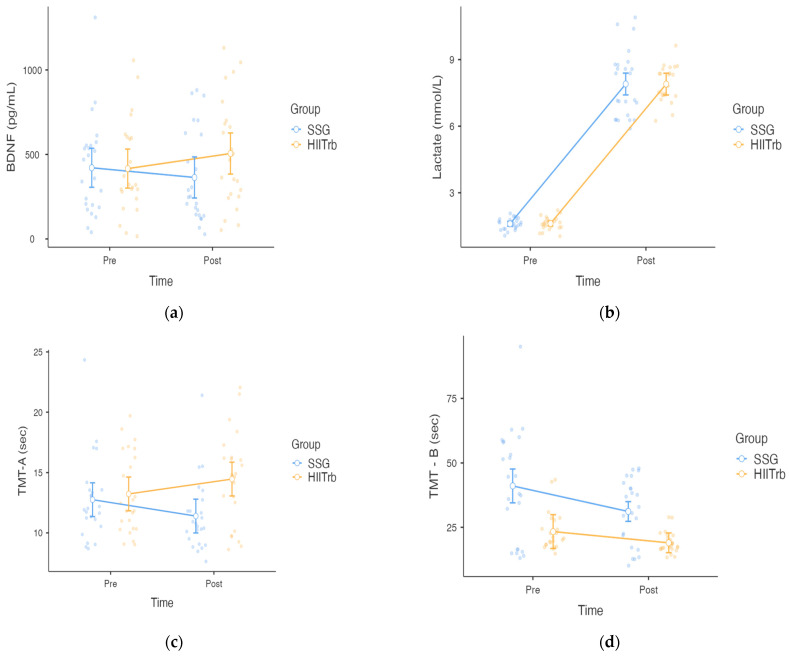
Neurochemical [serum brain-derived neurotrophic factor (BDNF) and lactate] and cognitive [Trail Making Tests A and B (TMT-A and TMT-B)] responses for SSG vs. HIITrb. (**a**) Serum BDNF levels measured before and after exercise; (**b**) lactate levels measured pre- and post-exercise. (**c**) TMT-A test scores (completion time) measured pre- and post-exercise. (**d**) TMT-B test scores (completion time) measured pre- and post-exercise. Error bars represent 95% confidence interval.

**Figure 3 healthcare-13-01738-f003:**
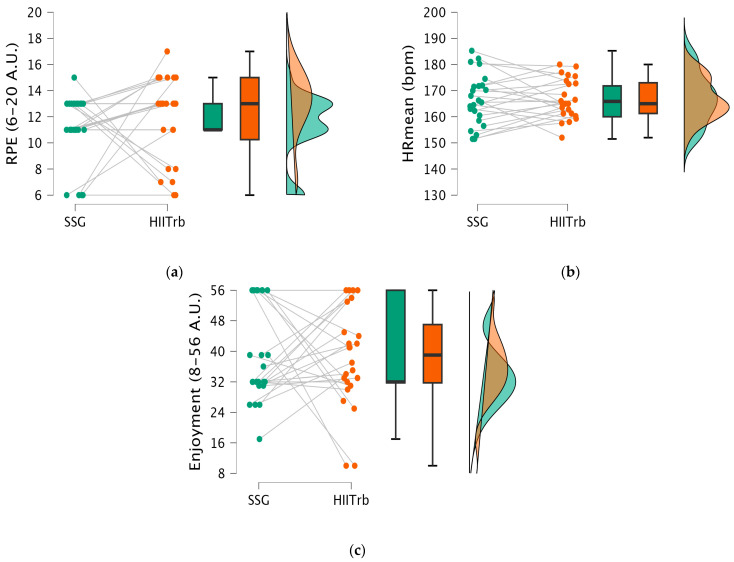
Psychophysiological responses (heart rate mean [HRmean] values, enjoyment score, and rating of perceived exertion [RPE] level) in the SSG vs. HIITrb condition. (**a**) RPE scores measured post-exercise; (**b**) HRmean (bpm) measured during exercise; (**c**) enjoyment scores measured post-exercise.

**Table 1 healthcare-13-01738-t001:** Characteristics of the small-sided game (SSG).

SSG	Coach Encouragement	No. of GK	Pitch Size (Length × Width)	Rule	No. × Duration of Series	Inter-Series Recovery	Total Duration
4-a-side	Yes	None/Small Goals	35 × 25 m	Free Play	4 × 4 min	3 min passive	~25 min

**Table 2 healthcare-13-01738-t002:** Characteristics of the running-based high-intensity interval training.

(No. × Duration of bout)	Work Velocity	No. × Duration of Series	Intra-Series Recovery	Inter-Series Recovery	Total Duration
8 × 15 s	110% MAS	4 × 4 min	15 sec Passive	3 min Passive	~25 min

MAS: Maximal aerobic speed.

**Table 3 healthcare-13-01738-t003:** Neurochemical and cognitive responses for SSG vs. HIITrb.

(*n* = 24)	SSG			HIITrb		
	Pre	Post	95% CI	Pre	Post	95% CI
	Mean ± SD	Mean ± SD		Mean ± SD	Mean ± SD	
BDNF (pg/mL)	421.07 ± 287.73	363.84 ± 275.75	304.14–480.77	416.16 ± 287.73	505.44 ± 315.26	372.49–549.11
Lactate (mmol/L)	1.60 ± 0.25	7.90 ± 1.49	4.50–5.00	1.62 ± 0.28	7.90 ± 0.79	4.50–5.01
TMT-A (s)	12.76 ± 3.49	11.40 ± 2.98	10.78–13.37	13.2 ± 3.30	14.46 ± 3.80	12.55–15.14
TMT-B (s)	41.08 ± 21.36	21.14 ± 12.56	31.14–41.07	23.34 ± 7.41	18.97 ± 4.05	16.19–26.12

SSG: small-sided game; HIITrb: running-based high-intensity interval training; SD: standard deviation; BDNF: brain-derived neurotrophic factor; TMT: Trail Making Test; Sec: second; pg/mL: picogram/milliliter; CI: confidence interval.

**Table 4 healthcare-13-01738-t004:** Psychophysiological responses to SSG and HIITrb conditions.

*n* = 24	SSG	HIIT(rb)	Mean Differences	%95 CI for Mean Differences
	Mean ± SD	Mean ± SD		
HRmean (bpm)	166. 59 ± 9.76	166.81 ± 7.51	−0.219	−3.843–3.406
RPE (6–20 A.U.)	11.17 ± 2.60	11.92 ± 3.26	−0.750	−2.672–1.172
Enjoyment (8–56 A.U.)	38.58 ± 12.32	38.46 ± 13.17	0.125	−8.421–8.671

A.U.: arbitrary unit; bpm.: beat per minute; HIITrb: running-based high-intensity interval training; HRmean: mean heart rate; SD: standard deviation; SSG: small-sided game; RPE: rating of perceived exertion; CI: confidence interval.

## Data Availability

The datasets generated and analyzed during this study are available from the Zenodo repository: https://doi.org/10.5281/zenodo.15924659 (accessed on 15 July 2025).

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
