# Peer review of "Acute Neurochemical, Psychophysiological, and Cognitive Responses to Small-Sided Games vs. Running-Based HIIT in Young, Male Soccer Players"

_healthcare, 2025, doi:10.3390/healthcare13141738_

Round 1

Reviewer 1 Report

Comments and Suggestions for Authors
  1. Basic Reporting

Strengths:

  • The manuscript is structured and follows standard scientific reporting.
  • The abstract concisely summarizes the aim, methods, results, and conclusions.
  • Appropriate ethical considerations and trial design are described.
  • Figures and tables align with the text and are appropriately referenced.

Suggestions for Improvement:

  • The manuscript would benefit from professional English editing. Several grammatical and stylistic issues hinder flow and clarity. For instance:

“A SSG was performed…” → should be “SSGs were performed...”

“players were informed to continue” → better as “players were instructed to maintain...”

  • The title could be made more concise and impactful. Suggested:

“Acute Cognitive and Neurochemical Effects of Small-Sided Games vs. Running-Based HIIT in Soccer Players”

  • Please clarify all abbreviations (e.g., SSG, HIITrb, TMT-A/B) when first used in each central section (Abstract, Methods, Results).
  • Some table/figure captions lack sufficient detail for standalone interpretation. Add explicit definitions (e.g., units, significance notation, sample size).
  • Use consistent statistical reporting: state sample size (n) with all mean ± SD values and ANOVA results.
  1. Experimental Design

Strengths:

  • The randomized crossover design is appropriate and strengthens internal validity.
  • Sample size estimation using G*Power is justified and meets power requirements.
  • All participants were from the same team, which increases control over training background and context.

Suggestions for Improvement:

  • The timing of BDNF measurement may affect results. Since BDNF levels fluctuate rapidly post-exercise, clarify why blood was drawn “immediately after” rather than with multiple post-exercise timepoints (e.g., 15 or 30 mins after).
  • Acknowledge that Trail Making Test A and B primarily assess processing speed and cognitive flexibility, respectively, and are limited in scope for broader executive function.
  • Could you explain why no control condition (e.g., rest or non-cognitive activity) was used for comparison?
  • Could you provide more details on participant familiarization with cognitive tests to minimize learning effects?
  • The use of only male athletes limits generalizability. We suggest acknowledging this and recommending follow-up studies with female athletes.
  1. Validity of the Findings

Strengths:

  • The comprehensive statistical analysis includes interaction effects, effect sizes, and post-hoc testing.
  • Both objective (HR, lactate) and subjective (RPE, enjoyment) psychophysiological measures are included.

Suggestions for Improvement:

  • The lack of change in BDNF is well-discussed, but the manuscript should clearly state this as a null finding, avoiding overinterpretation.
  • Consider discussing potential confounding from cortisol levels during the competitive season, which could suppress BDNF.
  • The finding that TMT-A improved but not TMT-B suggests a limitation in the scope of cognitive improvements. Discuss whether this reflects attentional rather than executive function benefits.
  • Although both protocols showed similar lactate responses, only SSG improved cognitive performance, highlighting that metabolic stress alone may not explain cognitive gains.
  1. Additional Comments

Suggestions:

  • Clarify how “enjoyment” was interpreted regarding training adherence or motivational implications.
  • The lack of novelty in enjoyment results might stem from players’ habituation—expand this in the limitations.
  • Encourage including longer-term studies to evaluate chronic adaptations and link them to performance outcomes.
  • Ensure data availability complies with FAIR principles. Consider publishing anonymized raw data to support transparency.
Comments on the Quality of English Language

The English could be improved.

Reviewer 2 Report

Comments and Suggestions for Authors

Please read carefully my suggestions in the attached file.

Title & Abstract
The title is original, clear and reflects the content of the study. In particular, the focus on comparing the neurochemical and cognitive effects of two different exercise models is positive in terms of the timeliness of the topic. The abstract is very structured; method, sample, results and conclusions are clear. However, the "coaches and practitioners" section could be a bit more detailed with a sentence.

Introduction
The introduction is supported by sufficient literature and the rationale and hypothesis of the study are well structured. However, the "open-skill" nature and cognitive benefits of SSG could be further emphasized. Also, previous studies on BDNF could have been summarized in a more systematic structure.

Methods
The research design (randomized, comparative, crossover model) is very strong. Participant characteristics were detailed. BDNF measurement timing, TMT test selection and exercise protocols are well described. However, a limitation is that the BDNF timing is only a single measurement after exercise. This should either be defended in the methods section or more clearly stated in the discussion section.

Results
Statistical analyses were conducted and reported correctly. Effect sizes (η²) are clearly presented. However, some of the tables and figures are too dense; therefore, simplification is recommended for visualization. The significant difference in TMT-A is impressive, but the lack of difference in TMT-B should be interpreted more clearly.

Discussion
The discussion section is well structured and consistent with the literature. Contradictory results on BDNF are explained in the light of the literature. However, more practical implications would increase the impact of the discussion. In addition, possible molecular mechanisms other than BDNF could be emphasized more.

Conclusion
The conclusion summarizes the findings but could be made somewhat more effective. In particular, it could include practical recommendations to answer the question "so what?" (e.g: "SSG formats may be preferable for cognitive stimulation in the short term").

Reviewer 3 Report

Comments and Suggestions for Authors

Peer Review: Acute Effects of Small-sided Games and Running-based High-Intensity Interval Training on Neurochemical, Psychophysiological, and Cognitive Responses in Soccer Players

  1. Initial Assessment
  • Scientific Contribution: The paper continues exercise-cognition research, by putting into question the acute-response on neurochemical, cognitive and psychophysiological parameters in small-sided games (SSG) versus running-based high-intensity interval training (HIITrb) in soccer players. It fills a new gap by building on well-known paradigms to connect exercise with cognitive benefits, and it focuses on a soccer-specific player-related framework by measuring BDNF-mediated effects.
  • Methodological Rigor: The proposed research was designed on the premises of a strict randomised, counterbalanced crossover design with a preconceived sample size of n=24 calculated in advance through power analysis. Despite the abovementioned strengths, several methodological drawbacks reduce credibility: the analytical error in the reporting results of cognitive data and the one-point approach to BDNF measurement. Moreover, although certain causes of bias are acknowledged ( e.g. no blinding of assessors), they are not compensated.
  • Theoretical Framework: This manuscript places itself in the modern debate about exercise-mediated cognitive enhancement from the perspective of brain-derived neurotrophic factor (BDNF). Despite such a match in directions, some of the most relevant findings pertaining to the lactate–BDNF interactions, as well as cortisol dynamics, have not been included in the review, which would have enriched the discussion on the same by providing a more harmonised overview of the existing evidence..
  • Global Relevance: This study will make a significant contribution to the global discourse because the cognitive advantage analysis after sport-specific training in a youth soccer environment helps to establish an effect that is generally helpful to different youth sports classes. However, the results can only be generalised since this is a study of a group of male youth soccer players.

Major Strengths:

  • The randomised crossover design minimises inter-individual variability, providing a strong framework for comparing the effects of SSG and HIITrb.
  • The multidimensional assessment (neurochemical, cognitive, psychophysiological) offers a comprehensive view of acute exercise impacts, as seen in the inclusion of BDNF, TMT, and enjoyment measures.

Principal Weaknesses:

  • A critical analytical flaw in the Trail Making Test (TMT) data reporting, with impossible baseline differences between conditions in a crossover design (e.g., TMT-A Pre-SSG: 18.65s vs. Pre-HIITrb: 12.76s), undermines the validity of cognitive performance conclusions.
  • The impossible ethics approval date (25 December 2024) raises serious credibility concerns, potentially leading to typographical errors or deeper compliance issues.
  1. Section-by-Section Assessment

Title

  • Evaluation: The suggested title is rather descriptive and correctly reflects the extent of the study, as it frames its primary variables (neurochemical, psychophysiological, cognitive), as well as the population to which it will be applied, soccer players. However, being 21 words, it is longer than the MDPI Healthcare requirement regarding the optimal length of the title that should be indexed, and it does not indicate the age and competition level of the population. The title should therefore not be very long, and it must be more concise hence the suggestion of having the following title: "Neurochemical and Psychophysiological Responses to Soccer Exercise in Adults: A Progressive Intensity Study"..
  • Recommendation: Minor revisions. Shorten and specify the population, e.g., "Acute Neurochemical, Psychophysiological, and Cognitive Responses to Small-Sided Games vs. Running-Based HIIT in Male Youth Soccer Players."

Abstract

  • Evaluation: This abstract follows the IMRAD format, briefly getting into the objectives of the study, its methods, results, and conclusions with relevant statistical information, and introduces a discrepancy in the range of the p-value of BDNF (p > .276 in Methods and p > .979 in Results). In addition, significant limitations, such as the lack of BDNF differences, were excluded. The report is too complex in terms of statistics, with F-values being able to be simplified in accordance with the requirements of the journal.
  • Recommendation: Minor Changes: The paper has been amended to eliminate gaps in statistics, streamline little statistical information (in the error omitting particular F-statistics), and include a small paragraph that defines possible weaknesses, thus performing clarity during the assumptions of hypotheses.

Introduction

The current research involved an in-depth literature review of soccer-specific physiological demands, the exercise-cognition interface, and mechanisms of BDNF, thus providing the research context. This is how the literature review develops a research gap or the investigation of neurobiological processes during small-sided games (SSG) as the main research question. However, the inference that there is no investigation that has looked into the molecular mechanisms which drive motor learning and executive functions is exaggerated by prior studies which have looked into soccer-specific BDNF responses. The correlation between SSG, BDNF, and cognitive performance can be understood by a well-articulated theory hypothesis, but its expression in the subsequent section is not in accordance with the MDPI standards, which advises the hypothesis to be presented in a separate paragraph. When put together, all of these problems require minimal changes to overcome a lack of methodological integrity and plausibility of the manuscript, possible with the insertion of neurochemical justification earlier, referencing of modern literature, and derivation of the hypothesis within the last, independent segment.

Methods

The aim of this study was to test the acute physiological and cognitive performance of two types of high-intensity exercise, sprint-specific gymnastics ( SSG ) and high-intensity interval running-based training ( HIITrb), using a randomised crossover design. This design was employed along with randomisation and counterbalancing; thus, a rigorous comparison was performed between the two exercises. However, several limitations of this study deserve particular attention

Study Design

A 7-days washout period was placed between exercise sessions; however, there are no data that can confirm such a long period of recovery. In addition, the possibility of carry-over effects was not explicitly mentioned or statistically considered

Participants and Recruitment

Although the statistical power of the sample is sufficient (24 male youth athletes recruited in one club offer more than 80 percent statistical power), it also limits generalisability. Details pertaining to the nature of recruited participants and the recruitment procedure by which they could have been achieved have not been stated, and in effect, questions relating to representativeness and the possibility of potential bias in selection cannot be ignored. 

Intervention and Outcome measures

Exercise protocols have been reported with sufficient specificity; measurement of brain-derived neurotrophic factor (BDNF) was restricted to one time point after exercise, essentially limiting the measurement of temporal variability. The test-retest reliability of the cognitive task, the Trail-Making Test, was not included, which can lead to underestimation of the stability of the performance and overestimation of the improvement connected to practice. In addition, there was no information on the BDNF ELISA coefficients of variation, which casts doubt on the accuracy of the assays

Statistical Approach

ANOVAs followed by repeated-measure analyses of variance and Bonferroni corrections were used correctly, but Greenhouse-Geisser 6 was not reported even when non-sphericity was present. There are only minor descriptions of how to deal with missing data and outliers, and there is no justification for why a parametric ANOVA is being used where skewness would have been anticipated

Recommendations

Several methodological concerns were identified: present empirical evidence to support the washout period, explain the mechanisms of blinding with regard to outcome measurements, describe the measure of precision of the BDNF assay, and provide data on the reliability of TMT to counteract the potential practice effects. To gain transparency, the authors can employ a conSORT flow diagram that would be in accordance with MDPI Healthcare policies.                           

Results

  • Evaluation: Results are clearly reported in an annalogicall mannerr addressing all preset results and reporting the statistics. However, there is a serious issue in reporting TMT baseline data ( observable pre-test differences in conditions ), which indicates a serious analytic mistake in a crossover design. Multiple confidence intervals are missing in the tables, identical numbers in figures (two Figure 1s), and such interpretative phrases (such as outperformed) should not be used in this section, according to the MDPI.
  • Statistical Reporting: Effect sizes have been reported in the statistical resultst,and there are inconsistencies in the p-values where the result,forexample.,the  BDNF group effect) have not been reported and the Greenhouse-Geisser 0valueshavetbeenn  reported.
  • Data Display: Table and figure representation are detailed but not polished,forexample., notations about vertical linegridsd in tables and lack of meaningfuls notations on a plot are inherent.
  • •Recommendation: Considerable revisions. Fix TMT analysis to denote within-subjects design, including confidence intervals in the tables, renumber a few figures, include notes describing the significant differences in the plots, deleting wording explanations, and identifying statistical inconsistencies..

Discussion

  • Interpretation of Findings: The interpretation of findings is situational, but the conclusion regarding cognitive performance is based on an erroneous analysis of thee TMT, so it has to be revised. Theorising the association between cortisol and muscle repair does not end with quantified data.

Contextualisation to Literature: The findings are compared to those of previous studies, yet the effects of competitive seasons on PDNF have not been studied thoroughly. Inconsistencies (e.g. null results of BDNF vs. positive changes) are noted, but cannot be determined completely.

  • Limitations:The limitationsdiscussed (for example, single-pointt BDNF measureise not very detailedabout assessorblindnessnand the, order effect. However, they do not provide easy navigation.
  • Implications: Practical implications of soccer training aredescribede althoughh the theoretical implications ofthe exerciseecognitionnmodeln are not outlinel.

Recommendation: Considerable revisions. Re-order cognitive interpretations after re-analysis of TMT indicate that less testing was done on those mechanisms that are speculative in nature, elaborate more on the effect of competitive seasons, and perhaps organise with clear subsections (for example, Findings and Implications) following MDPI Healthcare requirements.

Conclusion

  • Evaluation: The conclusion contains some of the crucial findingsthattares based on supposedly invalid TMT results on the grounds of errors in analysis. It does not have a short, citable formulation of the key contribution of the study which lacks an effective ending, as proposed by MDPI Healthcare.
  • •          Recommendation: Minor changes. Revise according to fixed TMT data and finish with a solitary declarative sentence (for example Game-based HIIT acutely improves processing speed in young male soccer players which is independent of circulating BDNF alterations)..

References

  • Evaluation: Reference list is quite thorough and the latest (until 2024), including the fields of physiology, neurochemistry, and cognition. Nevertheless, the problem is inconsistency in formatting (for example, DOI prefixes, journal abbreviations) is not following MDPI style. Older reviews are also redundant and high-impact 2024-2025 dose-response reviews of HIIT-BDNF are unavailable.
  • •          Recommendation: minor adjustments. Remove unnecessary citations, and insert new 3-5 high-impact reviews, and format to MDPI Healthcare guidelines.

Tables and Figures

  • Analysis: Tables and figures provide comprehensive data; however, their quality is rather poor: specific labels are doubled (e.g., two labels with the text Figure 1) and have no marginal notes of crucial differences. Tables contain vertical lines and fail to give confidence intervals going against MDPI formatting rules.
  • Recommendationsn: A few changes. Number figures consecutively and avoid vertical lines in tables, add confidence intervals and color-blind-safe color schemes with obvious significance labels on the plots.
  1. Ethical Considerations
  • Evaluation Ethicall approval and informedconsent weree obtainedin accordance withn the Declaration of Helsinki. However, the impossibility of an authorisation date (25 December 2024) is a serious issue. Information about the protection of the privacy of participants and the dangers of high-dose strategies was unavailable. The funding sources and conflicts of interest are fully disclosed.
  • Indication:Major revisionsr. Fix the date of approval, elaborate on privacy protection, and say something about the risks associated with exercise to comply with ethics in the reporting requirements standard of MDPI Healthcare.
  • .
  1. Statistical and Methodological Rigor
  • Evaluation: Statistical methods (repeated-measures ANOVA, Bonferroni corrections) are appropriate, but TMT data analysis appears to be fundamentally flawed with baseline differences in a crossover design. Multiple comparison adjustments are not fully reported in the results tables, and Greenhouse-Geisser ε values are omitted despite sphericity corrections. The handling of outliers or missing data is not discussed.
  • Recommendations: Major Revisions. Correct TMT analysis to reflect within-subjects design, report all statistical corrections explicitly, include ε values and adjusted degrees of freedom, and address the handling of missing data or outliers.
  1. Overall Evaluation

Major Strengths

  • A randomised repeated crossover research design was used to reduce inter-individual variability to allow facile analysis of the comparative effects of soccer-specific strength-endurance training (SSG) and high-intensity interval running training (HIITrb) on neurochemical, cognitive, and psychophysiological variables. A total of three variables (neurochemical, neurochemical-brain-derived neurotrophic factor, cognitive-trail making test, and psychophysiologicalexercise enjoyment and rating of perceived exertion) were assessed comprehensively. The introduction presents an in-depth literature review that puts the current research into the context of the available evidence. Reporting null results enhances methodological rigor as well as the body of knowledge. All these characteristics make the study empirically in depth and practically significant.

Critical Weaknesses

The reported TMT result contained an evident data error: there was an apparent difference between the baseline values during the crossover condition between the conditions, which could not have happened by chance. This disqualifies the cognitive findings of the study, hindering the need to conduct a thorough reanalysis.

 The provided date of ethics approval which is 25 December 2024 is beyond belief and presents a danger to compliance; the authenticity of the date must be verified.

 The concentration of BDNF was assessed only at one point, and there was therefore a limited interpretation, since these were not dynamic assessments.

Transparency is hampered by statistical inconsistencies; for example, the p-value of BDNF is not equal across the Abstract and Results as well as data reporting completeness. For example, Greenhouse-Geisser values have not yet been reported.

The alignment of tables and figures, such as the duplication of numbers and vertical grid lines, fails to meet the MDPI Healthcare formatting requirements and acts as a source of confusion..

Minor Issues

  • Terminology for participants varies (e.g., "amateur" vs. "sub-professional"), causing confusion and standardisation throughout.
  • Minor grammatical errors (e.g."strenous" ) and awkward phrasing reduce readability, and language editing is suggested.
  • Session duration inconsistency (25 vs. 45 minutes) between the Abstract and Methods needs clarification (active exercise vs. total session time).
